# A reversible dendrite-free high-areal-capacity lithium metal electrode

Hui Wang[1,†], Masaki Matsui[1,2,3], Hiroko Kuwata[1], Hidetoshi Sonoki[1], Yasuaki Matsuda[1], Xuefu Shang[1], Yasuo Takeda[1], Osamu Yamamoto[1] & Nobuyuki Imanishi[1]

Reversible dendrite-free low-areal-capacity lithium metal electrodes have recently been revived, because of their pivotal role in developing beyond lithium ion batteries. However, there have been no reports of reversible dendrite-free high-areal-capacity lithium metal electrodes. Here we report on a strategy to realize unprecedented stable cycling of lithium electrodeposition/stripping with a highly desirable areal-capacity ($12\,mAh\,cm^{-2}$) and exceptional Coulombic efficiency ($>99.98\%$) at high current densities ($>5\,mA\,cm^{-2}$) and ambient temperature using a diluted solvate ionic liquid. The essence of this strategy, that can drastically improve lithium electrodeposition kinetics by cyclic voltammetry premodulation, lies in the tailoring of the top solid-electrolyte interphase layer in a diluted solvate ionic liquid to facilitate a two-dimensional growth mode. We anticipate that this discovery could pave the way for developing reversible dendrite-free metal anodes for sustainable battery chemistries.

[1] Department of Chemistry for Materials, Faculty of Engineering, Mie University, Tsu 514-8507, Japan. [2] Department of Chemical Science and Engineering, Kobe University, 1-1 Rokkodai-cho, Nada-ku, Kobe 657-8501, Japan. [3] Precursory Research for Embryonic Science and Technology (PRESTO), Japan Science and Technology Agency (JST), 4-1-8 Hocho, Kawaguchi, Saitama 332-0012, Japan. † Present address: Energy and Environment Directorate, Pacific Northwest National Laboratory, 902 Battelle Boulevard, Richland, Washington 99354 USA. Correspondence and requests for materials should be addressed to H.W. (email: wang@chem.mie-u.ac.jp) or to N.I. (email: imanishi@chem.mie-u.ac.jp).

Dendritic growth of lithium deposits has plagued the reality of Li-metal-based batteries for approximate four decades, specifically in terms of safety and battery lifetime[1-3]. The persistent thrust to solve this issue has always been driven by the unprecedented characteristics of lithium electrode, that is, its high theoretical specific capacity of 3,861 mAh g$^{-1}$ and low electrochemical potential ($-3.04$ V versus standard hydrogen electrode), for the implementation of high specific energy Li-metal-based batteries. To circumvent the propagation of lithium dendrites, intense studies of metallic lithium electrodes (MLEs) have used strategies for conducting stable lithium electrodeposition such as maintaining a sustained supply of Li$^+$ in the vicinity of MLE surface[4-9], the spatial redistribution of Li$^+$ surge along tailored[10-14] or artificial solid-electrolyte interphase (SEI) films[15-18], enhanced Li$^+$ surface diffusivity[19,20] and the fabrication of Li metal with high surface energy[18,21,22]. As a result of these strategies, reversible dendrite-free low-areal-capacity MLEs (0.5-3.0 mAh cm$^{-2}$) have been developed. The underlying nature of these advances is closely related to the supply of a locally homogenous current density or the physical blocking of dendrite growth with less considerations of manipulating lithium electrodeposition kinetics itself.

High areal-capacity and excellent Coulombic efficiency (CE) of MLEs, dependent on the dendrite-free two-dimensional (2D) growth, are highly desirable for the development of high specific energy Li-metal-based batteries. In order for 2D growth mode to prevail, the interlayer transport of deposited lithium adatoms must be fast enough to prevent the onset of nucleation of islands on the yet undeveloped growing islands[23]. Kinetic models and physical arguments indicate that 2D growth should occur when the critical island size ($L_{crit}$) is greater than the separation between 2D nuclei ($L_s$). To fulfil this condition, enhancing the surface diffusivity of D and nucleation density (decreasing $L_s$) or reducing the magnitude of the Ehrlich–Schwoebel barrier ($E_{e-s}$) by the surfactant (increasing $L_{crit}$) are promising methods, which can be realized by the addition of halogenated salts[20] or enhancing the operating temperature[24]. Yang et al.[25] demonstrated an organic surface modifier to interact with the deposited metal by its functional organic group to yield a 2D growth mode. Based on the well-known mosaic SEI layer model, a possible candidate for a surfactant to decrease $E_{e-s}$ is oligomer species stemming from the solvent in the battery electrolytes that are spatially inhomogeneously distributed[26]. A unique, promising way of tailoring the top SEI layer, unveiled by Xu et al.[27,28], is the application of an electrochemically reduced solvation sheath (SS) of Li(solvent)$_n^+$ under the orientated migration, as long as the reduction potential of the neutral co-solvent falls in a narrow range. Recent advances on reversible Mg electrodeposition obviously manifest the crucial importance of solvent candidate (long-chain ether molecules) that has a high binding energy with Mg$^{2+}$ to create the robust SS[29-31].

Herein, we design and demonstrate a diluted solvate ionic liquid (DSIL) composed of the solvent of tetraethylene glycol dimethyl ether (G4 or C$_{10}$H$_{22}$O$_5$) and the salt of lithium bis(fluorosulfonyl)imide (LiFSI or LiN(SO$_2$F)$_2$) that possesses a robust SS of Li(2G4)$^+$ in the presence of the co-solvent 1,3-dioxolane (DOL or C$_3$H$_6$O$_2$) to realize high areal-capacity and superior CE of MLEs at ambient temperature. Various in situ or ex situ characterization and simulation tools have been employed to analyse the tailored top SEI layer and observe the growth mode of lithium electrodeposits aiming at correlating our proposed strategy with the 2D growth mode.

## Results

**Lithium electrodeposition onto Cu.** The co-solvent of DOL ($E_{h,DOL} = -1.48$ V versus Li/Li$^+$) was successfully screened with

attempts to decrease the viscosity of LiFSI-2G4 ($E_{h,G4} = -1.49$ V versus Li/Li$^+$)[32] on the premise of the preservation of the solvation manner of Li$^+$ at ambient temperature, as demonstrated by the dominant presence of solvating peak at 870 cm$^{-1}$ in spite of the added volume of DOL (Supplementary Fig. 1)[24]. The composition of LiFSI-2G4-50 vol% DOL was determined by a substantial increase in the ionic conductivity ($4.82 \times 10^{-3}$ S cm$^{-1}$), double increase in the lithium-ion transference number (0.63), drastic 10-fold decrease in the viscosity (2.98 mPa s), sweeping-dependent electrochemical window ($>5.5$ V after five sweeps) and reduced interfacial resistance (57 $\Omega$ cm$^2$ after 3 weeks) (Supplementary Figs 2 and 3), as compared with LiFSI-2G4. Cyclic voltammetry (CV) measurements reveal unexpected increases of plating current density ($I_p$) using the solvate ionic liquid (SIL) of LiFSI-2G4 and the DSIL of LiFSI-2G4-50 vol% DOL in Fig. 1a. After 30 cycles, the onset potential of bulk lithium electrodeposition onto Cu substrate was shifted positively for the SIL and the DSIL; the overpotential for $I_p$ of 5.0 mA cm$^{-2}$ was $-0.16$ V for the SIL and $-0.07$ V for the DSIL as shown in Fig. 1b and Supplementary Fig. 4. Energy dispersive X-ray Cu mapping (purple) of the Cu electrode after CV measurements (premodulation) shows that lithium nuclei were distributed on the Cu surface, some of which are coalesced into a large planar lithium island of 10-30 μm in length with concentrated O elemental mapping in orange (Supplementary Fig. 5). The surface chemistry of MLE after CV premodulation does not initially exhibit the hard-removed inherent contamination from lithium carbides (Li$_x$C$_y$) species[33,34], however, after 5 min of etching, they instantaneously appear beneath a G4-enriched top-SEI film from the SS of Li(2G4)$^+$ (Supplementary Fig. 6). It is reasonable to assume that a non-thermodynamic-reduced SEI chemistry will take place on the surface of newly formed lithium islands and the uncovered lithium metal substrate, as a result of the preferential reduction of the SS over that of DOL (the decreased decomposition from DOL at the top surface as shown in Fig. 5b)[28].

Morphology evolution of lithium electrodeposits with deposition time was monitored by a home-built visualization Li|Cu cell (see Methods section) as shown in Fig. 1c. Figure 1d and Supplementary Movie 1 show typical images of lithium electrodeposits in a conventional carbonate-based electrolyte at 1.0 mA cm$^{-2}$ and ambient temperature. In agreement with commonly accepted knowledge, three-dimensional un-controllable propagation of lithium electrodeposits takes place onto the top of randomly distributed lithium nuclei. However, the distribution of lithium nuclei was uniformed in the case of DSIL as shown in Fig. 1e and lithium electrodeposits with a high spatially homogenization were grown along the spherical Cu surface with time to totally cover Cu surface at 120 min (Supplementary Fig. 7a). After the application of CV premodulation, the distribution of lithium nuclei was even more uniform and a full coverage of Cu was achieved more rapidly. Lithium electrodeposition was observed to grow coaxially and layer by layer as shown in Fig. 1f and Supplementary Movie 2. Even at 5.0 mA cm$^{-2}$, the same 2D layer-by-layer dendrite-free growth mode was witnessed in Supplementary Fig. 7c and Supplementary Movie 3. However, inhomogeneous and non-coaxial lithium electrodeposition was observed in other electrolytes as shown in Supplementary Fig. 7b,d–f.

**Cycling stability of MLE.** CE of the Li|Cu cell using the DSIL was found to be strongly dependent on the CV premodulation. In Fig. 2a, the initial CE of the cell was drastically enhanced from 39.0 to 90.5% at 5.0 mA cm$^{-2}$ and 3.0 mAh cm$^{-2}$ upon the application of the CV premodulation, where the new calculation

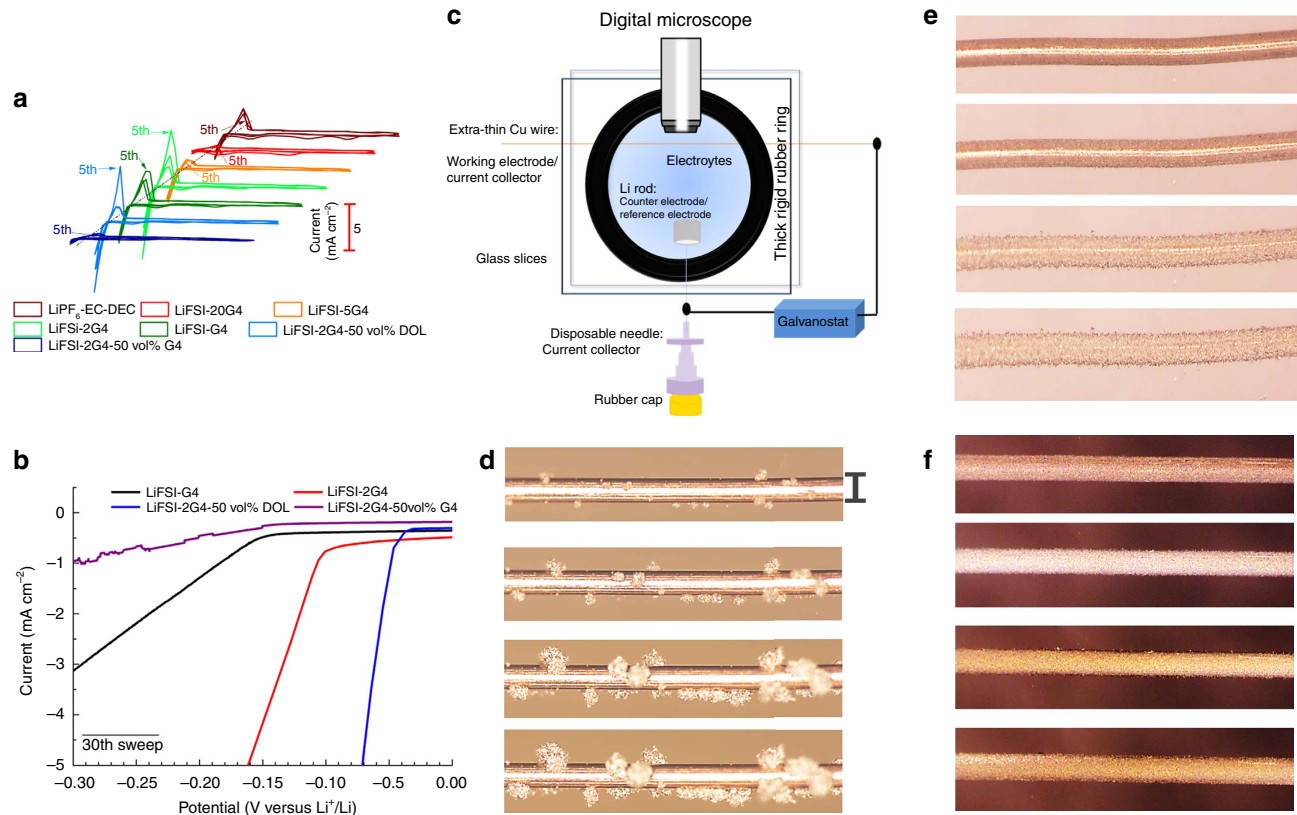

**Figure 1 | Lithium electrodeposition onto Cu. (a)** Cyclic voltammograms of Li electrodeposition/stripping in various electrolytes using Cu as the working electrode and Li as the reference and counter electrode for five cycles. (**b**) The electrodeposition behaviour of lithium in the 30th sweep. (**c**) Schematic representation of *in situ* visualization cell for observing the morphology of lithium electrodeposits. (**d–f**) Deposition time-dependent (2.5, 10, 60 and 90 min) morphology evolution of lithium electrodeposits at 1 mA cm$^{-2}$ and ambient temperature in the (**d**) LiPF$_6$-EC-DEC, (**e**) LiFSI-2G4-50 vol% DOL and (**f**) LiFSI-2G4-50 vol% DOL after the CV premodulation, respectively. Scale bar, 100 μm.

method (equation 1) takes into account of the $Q_r$—the capacity for residual lithium electrodeposits onto Cu substrate after the CV premodulation. As shown in Supplementary Fig. 8, the $Q_r$ was determined as 0.25 mAh cm$^{-2}$. In Fig. 2a, compared to symmetric lithium plating/stripping in the Li|Cu cell without CV premodulation, distinct difference in the cell overpotential was observed for the Li|Cu cell under the CV premodulation, where the stripping overpotential of 32 mV is much lower than the continuously increasing plating overpotential from 130 to 190 mV, indicating continuous nucleation of lithium electrodeposits occurs on the Cu substrate upon plating and the other side of Li electrode renders lithium electrodeposition in a more facile path. Moreover, the initial CE of the cell surprisingly increases with increasing fixed plating areal capacity in Fig. 2b. The calculated initial CE is 38.4% for 1 mAh cm$^{-2}$, 90.5% for 3 mAh cm$^{-2}$, 95.2% for 5 mAh cm$^{-2}$ and 98.0% for 12 mAh cm$^{-2}$. For long-term cycling tests, the average CE of the cell delivering 3 mAh cm$^{-2}$ is 83.2% for 8 cycles, 99.73% for 500 cycles and 99.86% for 999 cycles in Fig. 2c. From the eighth cycle, the theoretical stripping areal capacity of 3 mAh cm$^{-2}$ can be realized; the average CE of the cell delivering 12 mAh cm$^{-2}$ was calculated to be higher than 99.98% for 450 cycles in Fig. 2d. Two typical plating plateaus were always witnessed in Fig. 2c,d, where the second plateau with higher-plating overpotential regularly follows the first plateau with lower-plating overpotential and the stripping capacity is consistently higher than the plating capacity from the first plateau in each cycle but the overpotential between these two reverse processes is symmetrical. Figure 2e,f show excellent cycling stability of the Li|Li cell delivering the areal capacity of 12 mAh cm$^{-2}$ with the DSIL after CV premodulation

at 5.0 mA cm$^{-2}$ and 25 °C for 100 cycles (total-charge passed: 8,640 C cm$^{-2}$). No sudden decrease, spikes and erratic fluctuations in the cell potential were observed on cycling. Two distinct plateaus were also observed in the Li|Li cell. Scanning electron microscopy (SEM) top-view and cross-sectional views of lithium electrodes after cycling reveal dendrite-free planar lithium surface with pits and laminated lithium planar films (Supplementary Fig. 9). EDX mapping from the cross-sectional electrodeposited Li presents the concentrated atomic ratio of C and O probably stemming from the SS of Li(2G4)$^+$ (Supplementary Fig. 10)[27]. Even at an exceptionally high current density of 10.0 mA cm$^{-2}$, stable cycling was also realized under the same conditions (Supplementary Fig. 11). To the best of our knowledge, this is the first report of a symmetric coin-cell of Li|Li showing superior cycling with a practical high areal-capacity of 12 mAh cm$^{-2}$ at a high current density of 10 mA cm$^{-2}$ and 25 °C.

**Sweeping time-dependent positively charged lithium substrate.** To elaborate the key role of CV premodulation, CV measurements were performed under different conditions in Fig. 3a. CE of lithium electrodeposition was found to strongly rely on the sweep rate and potential range. During the first cathodic sweep at 5.0 mV s$^{-1}$, the lithium stripping rate from the lithium is faster than the lithium electrodeposition rate on the Cu substrate (0 to −0.3V). The lithium plating rate increased in the range of −0.3 to −0.2 V, possibly due to the change of deposition sites from the Cu–Li interface to Li–Li interface on Cu in Fig. 3c. A substantial amount of lithium stripped from the lithium anode remained in the bulk electrolyte and participated in the reverse step of lithium electrodeposition onto lithium cathode to render

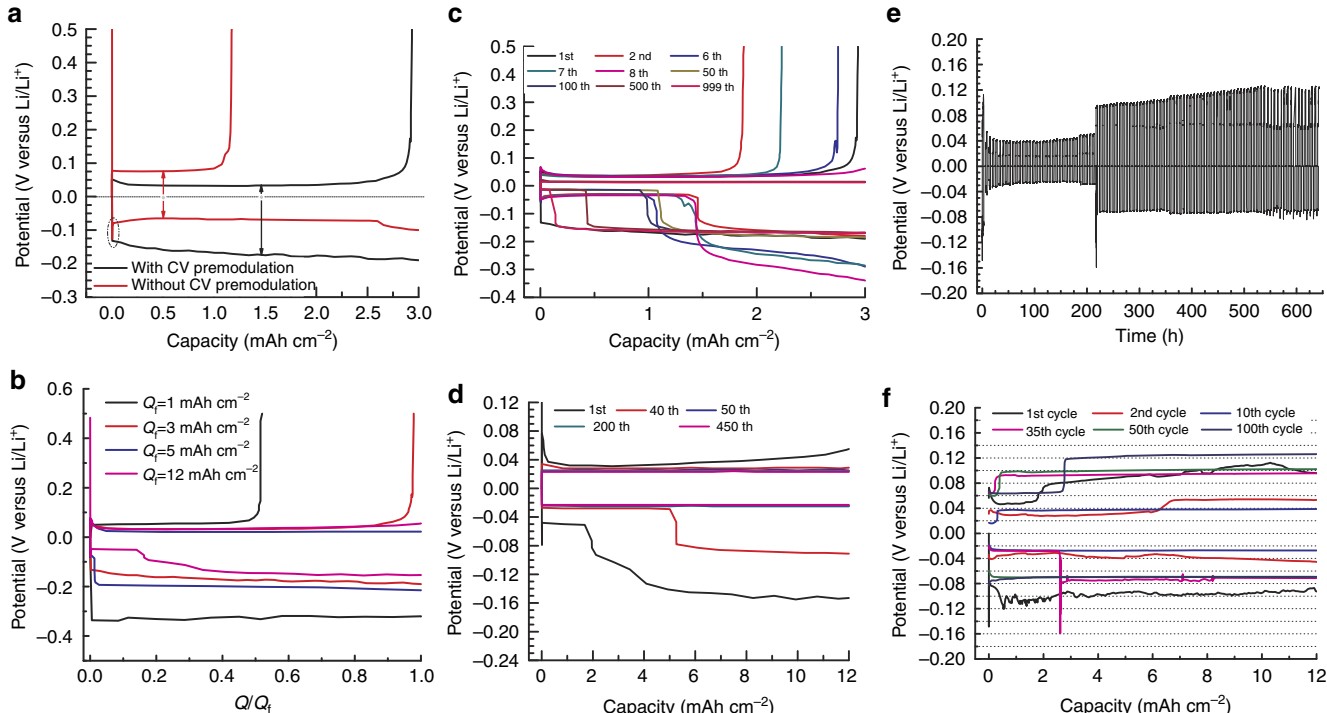

**Figure 2 | Cycling stability of MLE.** Electrochemical performance of lithium electrodeposition/stripping on Cu or Li electrode in the DSIL of LiFSI-2G4-50 vol% DOL at 5.0 mA cm$^{-2}$ and 25 °C. (**a**) Impact of CV premodulation on the first CE. (**b**) Dependence of the fixed plating capacity of Li|Cu cells on the first CE after CV premodulation; (**c,d**) voltage versus capacity profile for the Li|Cu cell under CV premodulation with a fixed plating capacity of 3.0 mAh cm$^{-2}$ (**c**) and 12 mAh cm$^{-2}$ (**d**); voltage versus time (**e**) and voltage versus capacity (**f**) profile for Li|Li cells after CV premodulation with an areal capacity of 12 mAh cm$^{-2}$.

CE greater than 100% at 5.0 mV s$^{-1}$. During the first anodic sweep, lithium electrodeposition onto lithium cathodes is observed to be facilitated at low sweep rate (that is, the longer span for creating positively charge lithium anodes to adsorb multidentate solvent of G4 and then facilitate lithium electrodeposition kinetics) in Fig. 3b. At a higher sweep rate of 50 mV s$^{-1}$, the similar evolution was usually witnessed at the fourth sweep. During the second cathodic sweep at 5.0 mV s$^{-1}$, the plating current density for lithium electrodeposition onto Cu is comparable to the first stripping current density but twice higher than the first plating current density in Fig. 3c. Lithium electrodeposition takes place onto previous residual lithium electrodeposits in the ensuing cathodic sweeps. As a result, the Li|Cu cells should be regarded as the Li–Li|Cu–Li cells from the second sweep at 5.0 mV s$^{-1}$.

During the cathodic sweep (3.0–0.6 V), the abnormal increase in the charge was monitored in the 30th sweep (Supplementary Fig. 12a). The lithium stripping from Cu–Li occurs even during the cathodic sweep. Regardless of the cathodic sweep or the anodic sweep in the range of 3.0–0.6 V, the Cu–Li is always positively charged to allow lithium stripping from Cu to take place. After 30 sweeps, the CE gradually decreases to 100% as a result of the substrate transformation from pristine Cu into Cu–Li. A large fraction of lithium stripping process occurs in the range of 2.2–3.0 V after the anodic sweep potential range changed from 0–0.9 V to 0–3.0 V owing to the dependence of the availability of positively charged lithium residuals onto Cu electrode on the high positive potential (Supplementary Fig. 12b).

After CV premodulation, galvanostatic lithium electrodeposition was performed on Cu substrate for a short period to observe the morphology of lithium electrodeposition by SEM as shown in Fig. 3d. Two distinct plateaus were again produced. Large planar patch-like lithium electrodeposits with ca. 30 μm in width were

observed to grown onto the smaller lithium particles generated from the CV premodulation and to coalesce with neighbouring analogues.

To further study the impact of CV premodulation ahead of the bulk lithium electrodeposition, *in situ* CV/quartz crystal microbalance (QCM) measurements were performed using three-electrode cells in the range of 0–3.1 V in different electrolytes. No increase in the peak current was observed in the electrolyte of LiPF$_6$-EC-DEC (Supplementary Fig. 13a). The peaks during the cathodic and anodic sweeps are comparable for SIL and DSIL (Supplementary Fig. 13b,c) with an unexpected increase in the current density that are ascribed to underpotential lithium deposition/stripping[35]. During the second anodic sweep, the continuous increase in the mass of the Ni substrate using DSIL presents solid evidence that G4 is supposed to interact with positively charged lithium deposits derived from the underpotential deposition during the second cathodic sweep at 0.95 V in Fig. 3e. However, the increase in the mass of the Ni substrate during the second anodic sweep for SIL was not observed, which suggests that G4 is not facile to interact with the positively charged lithium deposits in the high viscous SIL. After four sweeps, Ni-coated quartz crystal has been coated by underpotential-deposited lithium. From 3.0 to 1.5 V, the slight decrease in the mass implies de-adsorption of G4 from positively charged lithium on Ni (Supplementary Fig. 13c). The irreversible mass of $\Delta m_{irr}$ upon cycling, referring to the accumulation of SEI, was observed to be negligible for the DSIL electrolyte when the cycling number increased to five (Supplementary Fig. 13d), but continuously increased for the SIL and carbonate-based electrolytes. According to the slope of the two curves arising from the potential zone of 1.3 to 0.42 V and 1.3 to 2.65 V[36], the molar mass of the species for electrodeposition/stripping was calculated as 6.94 g mol$^{-1}$ that is close to that of Li (Supplementary

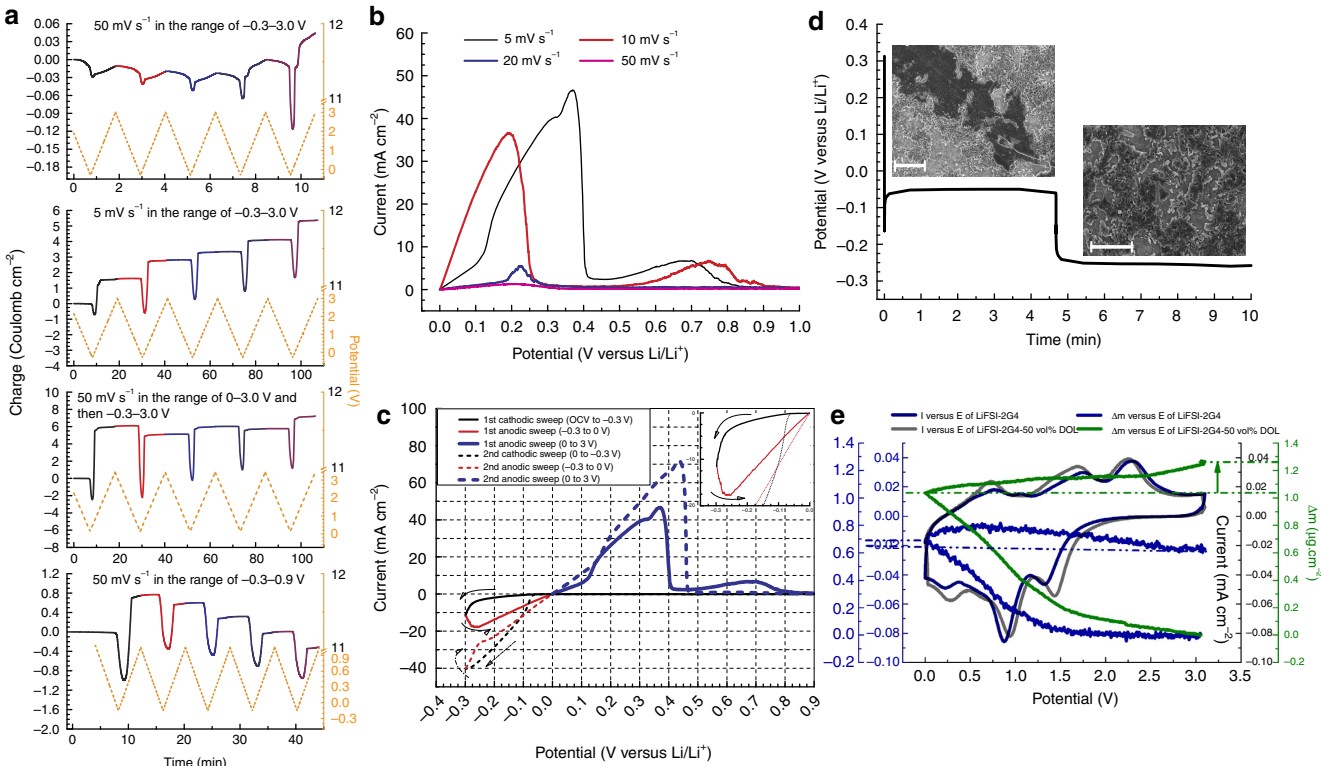

**Figure 3 | Reversible facilitating lithium electrodeposition by positively charged lithium substrate.** (**a**) Dependence of sweep rate and potential range on the CE of lithium electrodeposition during CV measurements of a Li|Cu cell at 25 °C. (**b**) The first anodic sweep at different sweep rates (**c**) the first two cycles of Li|Cu cell at 5.0 mV s$^{-1}$ and 25 °C in the range of − 0.3–3.0 V (**d**) two plateaus of lithium electrodeposition onto Cu electrode with CV premodulation at 5.0 mA cm$^{-2}$, 3 C cm$^{-2}$ and 25 °C along with two typical images of morphology of lithium electrodeposits at different height positions. Scale bar, 20 μm. (**e**) *In situ* CV/QCM study on the underpotential Li deposition/stripping in the second cycle using three-electrode cell, where Ni-coated quartz crystal was used as the working electrode and Li as the reference and counter electrode, respectively.

Fig. 13e,f). The underpotential lithium electrodeposition/ stripping are further demonstrated. On the Cu-coated quartz crystal, the resembling cathodic peak for lithium underpotential deposition was observed at 1.48 V and the anodic peak for lithium underpotential stripping was observed at 2.57 V (Supplementary Fig. 14a); the cycling number for generating stable SEI was found to be 15 (Supplementary Fig. 14b). The decline in the OCV of the Li|Cu cell after 20 cycles represents a modification of the Cu surface (Supplementary Fig. 14c). Patch-like lithium electrodeposits and large dendrite-free lithium particles were both observed on Cu substrate (Supplementary Fig. 14d,e)[37,38]. After the galvanostatic deposition of lithium at 5.0 mA cm$^{-2}$ and 3.0 C cm$^{-2}$ with CV premodulation (Fig. 3d), patch-like lithium electrodeposits onto Cu were still present (Supplementary Fig. 14f).

**Impact of CV premodulation.** *In situ* Li|Cu visualization cell using the DSIL was also used to record the morphology evolution in the first two cycles at 5.0 mA cm$^{-2}$ and 1.5 mAh cm$^{-2}$ and ambient temperature under the CV premodulation as shown in Fig. 4a and Supplementary Movie 4. Upon the second plating, the potential declined slowly to reach the bulk deposition zone (< 0 V), indicating underpotential lithium electrodeposition took place in the galvanostatic deposition mode in the presence of positively charged lithium residuals with adsorbed multidentate G4 solvents. Upon the second stripping, the appearance of black SEI layers changed from the previous random way into a mossy-film-growing mode before 0.5 V, which may be associated with ensuing 100% CE in Fig. 2c. A symmetric cell of Li|Li was applied by CV premodulation in the potential range of − 0.3–0.3 V and

the impedance evolution of the cell was real-time monitored with cycling as shown in Fig. 4b–d. Compared to the cell with LiPF$_6$-EC-DEC, the cell with the DSIL exhibits drastic increases in the lithium plating and stripping current density in the fourth sweep and substantial decreases in the two depressed semicircles at the high frequency region from SEI-film resistance ($R_f$) and at the low frequency region from charge-transfer resistance ($R_c$). Two typical impedance spectra were fitted using the equivalent circuit model as shown in Supplementary Fig. 15a,b. After 10 cycles, $R_c$ was surprisingly found to have a trivial contribution to the whole spectrum with a five-fold declined $R_f$, probably due to the sharp increase in the effective surface area ($S_e$). After CV premodulation, the $R_c$ gradually recovered to the initial value but $R_f$ only recovered to a third of the initial value after 1 week (Fig. 4e and Supplementary Fig. 15c), which further demonstrates that $S_e$ had been enlarged and accustomed stable $R_{ct}$ can be tuned by our strategy of CV premodulation (Supplementary Fig. 15d).

**Favourable surface film chemistry.** As to the surface chemistry of Li or Cu electrodes, the elemental depth profile of the Cu substrate after CV premodulation was found akin to the top-SEI film of Li metal surface in Fig. 5a. The reduction species of $(CH_2CH_2OCH_2O-)_n$ and $(CH_2CH_2O-R)_n$ from reduced DOL, possessing two characteristic peaks at 288.6 eV and 286.2 eV, were found in the outmost SEI layer (0–1 min), respectively[39,40]. For the F 1s spectra, the majority of LiF is detected at the outermost surface (10–20 s), compared to the continuing increase contribution of LiF with depth in the case of LiPF$_6$-EC-DEC and LiFSI-2G4 (Supplementary Fig. 16). The Li 1s spectra distinctly display that Li$_2$O is the main constituent of the inner-SEI film

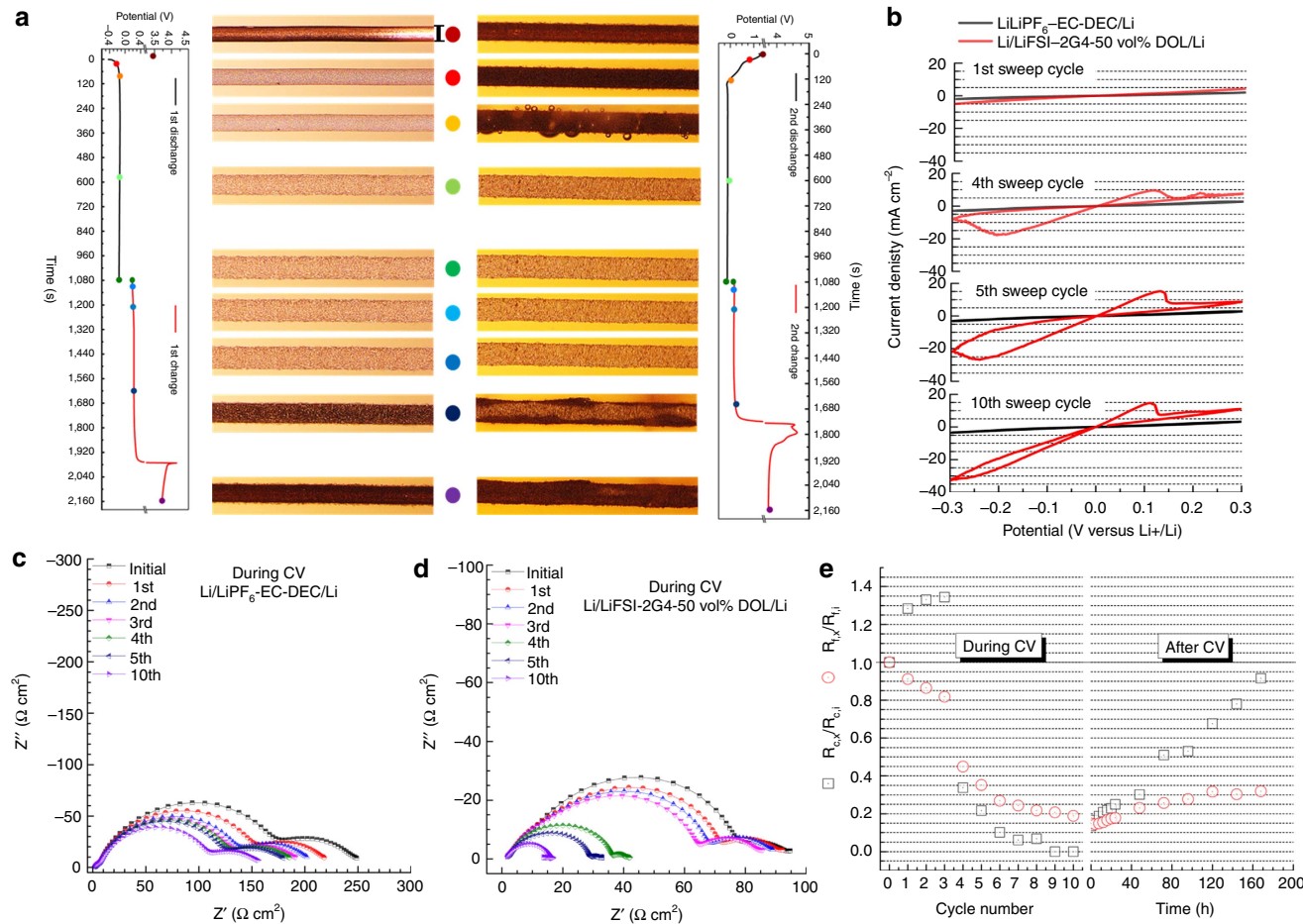

**Figure 4 | Impact of CV premodulation.** (**a**) *In situ* morphology evolution and chronopotentiometric curves of Li|Cu cell using the LiFSI-2G4-50 vol% DOL at 5 mA cm$^{-2}$ and ambient temperature under the CV premodulation in the first two cycles, where lithium electrodeposition and stripping time were fixed as 18 min (1.5 mAh cm$^{-2}$). Scale bar, 100 μm. (**b**) CV of Li electrodeposition/stripping in the symmetrical Li|Li cell for 10 cycles using different electrolytes at 25 °C. The sweeping rate was 10 mV s$^{-1}$. Impedance evolution of the symmetrical Li|Li cell with cycle number using (**c**) LiPF$_6$-EC-DEC and (**d**) LiFSI-2G4-50 vol% DOL. (**e**) Evolutions of $R_{c,x}$ and $R_{f,x}$ for the symmetrical Li|Li cell using LiFSI-2G4-50 vol% DOL during and after the CV premodulation.

(20–40 min) with little contribution from Li$_2$S and Li$_2$CO$_3$/LiOH mainly comprises the top-SEI film with small fraction of LiF. O 1s spectra present a clear transition after 5 min etching as well; the main inner contribution of Li$_2$O from the FSI(-F) is further determined. However, negative shifts of O 1s and Li 1s spectra to Li$_2$O were not observed in the case of LiFSI-2G4. The S 2p spectra show the reduction species of FSI(-F) at 167 eV constitutes the top-SEI film (0–5 min), demonstrated by the presence of the N 1s spectra featured at 399 cm$^{-1}$ (Supplementary Fig. 16). Without the addition of DOL, the reduction of FSI$^-$ to FSI(-F) and LiF scales with the depth. However, the addition of DOL could exhaust the FSI(-F) anion radicals and trigger the further reduction of FSI(-F) into the favourable generation of Li$_2$S$_2$ and Li$_2$S and the resulting products of FSI(-F)-S-S-FSI(-F)[41] characteristic of the N 1s peak at 402.6 eV (Supplementary Fig. 17), and LiF by the spontaneous dimerization reaction between FSI(-F) and Li$_2$S$_2$. CV premodulation could also impart the complete coverage of MLE by the new top-SEI film stemming from the SS in the DSIL regardless of the presence of the contaminant of Li$_x$C$_y$ as illustrated in Fig. 5c.

## Discussion

The strong interaction between G4-derived top-SEI film and the incoming Li cations could retard the surge of lithium ions during the nucleation stage[15,42] (Supplementary Figs 18 and 19). However, the lower-binding energy from the weak metallic bond of Li–Li or Li–Cu cannot inhibit the occurrence of local lithium surges. Besides, compared to surface diffusion on a terrace, there is a higher activation energy barrier of $E_{e-s}$ for an adatom to move over a descending step[23]. The preferred location of LiF in the top-SEI layer could enhance the surface diffusivity[20]; the G4-derived top-SEI film, by thermodynamic reduction and preferential electrochemical reduction from the SS, to function as the surfactant to reduce $E_{e-s}$ and then increase $L_{crit}$, especially by the redistribution of the dipolar SS of Li(2G4)$^+$ toward the step-edge region with the focused electrical field (Supplementary Fig. 20). With CV premodulation (reducing $L_s$), a high density of pre-existed 2D patches is benign to suppress lithium dendrite formation in the ensuing electrodeposition with a high deposition capacity (Supplementary Figs 5b and 14d).

Two distinct plateaus of Li|Cu cells and Li|Li cells in Fig. 2 indicate the first lower-overpotential plateau should be associated with the lithium electrodeposition onto top of residual lithium electrodeposits in a growing fashion of dendrite-free 2D large planar lithium particles at higher positions (Fig. 3d); the higher overpotential plateau could be ascribed to the lithium electro-deposition onto pristine Cu surface or bottom lithium surface directly at lower positions. One can see the lithium stripping capacity is consistently higher than the lithium plating capacity

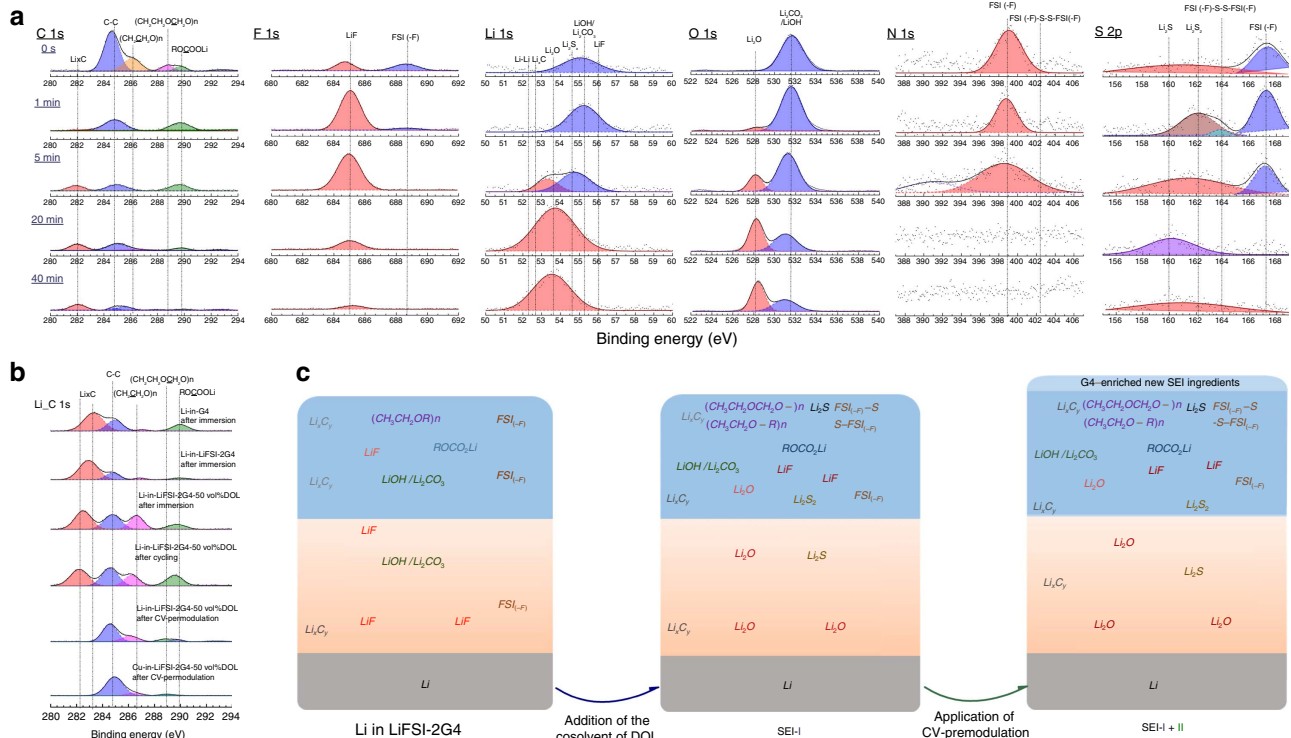

**Figure 5 | Favourable surface film chemistry of Li in the DSIL with the CV premodulation.** (**a**) Elemental depth profile of surface film chemistry of Li after CV premodulation in the DSIL of LiFSI-2G4-50 vol% DOL. (**b**) C 1s spectra of Li immersing in different electrolytes without etching. (**c**) Schemes of the modification of surface film chemistry of Li in the SIL of LiFSI-2G4 after the introduction of DOL and CV premodulation.

originating from the first plateau, indicating that dendrite-free large planar lithium particles are pivotal for complete lithium stripping. Moreover, the higher overpotential plateau regularly follows the lower overpotential plateau, suggesting 2D layer-by-layer growth mode prevails in our experiments evidenced by the continuously increasing plating overpotential and otherwise three-dimensional lithium dendrite growth is expected to occur on the top surface of lithium electrodeposits instantly with a shorter electrodeposition path. The fraction of the second plateau for cells delivering high areal-capacity ($>5\,\mathrm{mAh\,cm^{-2}}$) is so large that the lithium electrodeposition on Cu substrate at lower spots have sufficient time to grow till the coalescence with the top lithium electrodeposition layer within the unidirectional galvanostatic plating process. However, in the case of low areal-capacity, the arrival time for the coalescence of these two layers should be substantially delayed (after eight cycles for the cell delivering $3\,\mathrm{mAh\,cm^{-2}}$) to result in the initial fluctuations of CE.

Favourable surface chemistries of Li in contact with DSIL, on the grounds of the outmost layer of oligomers from DOL, the descending top-layer of LiF, $Li_2S_2$ and $Li_2S$, the inner-layer of $Li_2O$, have been reported to enhance the cycling stability of metallic lithium deposition[11,39,40,43]. The final negligible $\Delta m_{irr}$ upon cycling indicates a stable SEI film can be generated onto Li metal in contact with DSIL that leads to excellent cycleability of Li|Li cells with steady cell potentials.

In summary, we have reported evidences that CV premodulation is a viable strategy to generate a homogenous top-SEI cloth by preferential electrochemical reduction of the SS to fully cover MLEs with a low $E_{e-s}$ and to enhance the 2D growth mode in the presence of sweeping time-dependent positively charged lithium using the novel DSIL. DOL plays an important role in generating preferably localized SEI gradients of MLEs to sustain the excellent cycling performance. Reversible MLEs can then deliver an exceptionally high areal-capacity of $12\,\mathrm{mAh\,cm^{-2}}$ at high-current densities and ambient temperature. With the CV premodulation, the couple of Li/DSIL could be an appealing alternative for beyond lithium ion batteries. The proposed strategy could also be imposed to develop other reversible dendrite-free high-areal-capacity metallic anodes (for example, Na, Mg, Zn and Al) for sustainable battery chemistries.

## Methods

**Preparation of diluted solvate ionic liquids.** G4 (LBG, Kishida Chemical) and LiFSI (LBG, Kishida Chemical) were mixed in different molar ratios in an Ar-filled glove box. The mixtures of LiFSI-xG4 (x: the molar ratio of G4 to LiFSI) were stirred for 24 h at room temperature and the as-prepared homogenous electrolytes were stored and handled in the glove box. Blends of 1,3-dioxolane (DOL, 99%, Sigma-Aldrich) and LiFSI-2G4 were then combined using a pipette (Gilson Inc.) and stirred for an additional 24 h. The final residual moisture in all mixtures was determined to be $<20$ p.p.m., measured with a Karl Fischer moisture metre (KEM, MKC-510). The 1 M LiPF$_6$-EC-DEC (Kishida Chemical, 1:1 v/v) electrolyte was used for comparison. The other lithium salt evaluated in this work was lithium bis(trifluoromethanesulfonyl)imide (LiTFSI). LiTFSI (98%) was obtained from Wako Chemicals and dried overnight under vacuum at 150 °C. The washing solvent of 1,2-dimethoxyethane-DME (LBG, Kishida Chemical) and dimethyl carbonate-DMC (anhydrous, $>99\%$, Sigma-Aldrich) was used to remove the residual electrolyte of glyme-lithium salt mixtures and LiPF$_6$-EC-DEC on Li or Cu electrodes, respectively.

**Cell assembly.** *In situ* visualization Li|Cu cell: An extra-thin Cu wire (99.9%, Nilaco Corp.) with a diameter of 100 μm penetrated through a rigid-thick rubber ring, whose inner part (8–12 mm in length; 0.025–0.038 cm² in surface area) was employed as the working electrode for the lithium electrodeposition and outer part was used as the current collector. A disposable needle was then inserted into the rubber ring in the vertical direction, whose inner part was amounted by a Li rod using as the counter electrode and reference electrode and outer part acted by the current collector. The rubber ring was then sandwiched by two glass slices and injected by the electrolytes in the glove box. Electrodeposition of lithium onto Cu wire was carried out at ambient temperature (25 ± 2 °C) by an automatic polarization system (Hokuto Denko, HSV-110) using the current range of 10 μA. *In situ* observation of evolution of lithium electrodeposits with time was realized by a digital microscope (Keyence VHX-1000).

Coin-cell: Li|Li or Li|Cu coin cells were constructed from CR2032 stainless steel parts. The CR2032 cell is assembled with a steel cap incorporating a rubber sealing gasket. The contents include lithium foil wetted with the electrolyte, a nano-porous sheet separator nano-porous sheet separator (NPSS, 200 nm of mean pore size, 040A2, NSG group) soaked with electrolyte, and a second lithium foil or copper foil. A steel spacer acts as a current collector and is placed on top of this laminated electrode to inhibit contact between the steel spring and lithium foil. Finally, a cap is placed onto the entire assembly, which is then transferred to hydraulic crimper (Hoshen 2032). The lithium electrodes used (Honjo Metal) were 0.2 mm thick foil cut into 14 mm diameter disks. The average weight of the lithium electrode was 16.6 mg, which corresponds to the theoretical-specific areal-capacity of 41.6 mAh cm$^{-2}$. The NPSS used was 16 mm in diameter. All of the cell parts were dried in a vacuum oven at 80 °C for 24 h before use.

**Instrumentation.** Li electrodes immersed in different electrolytes were thoroughly washed by solvent using a Vortex-Genie 2 shaker (Scientific Industries, Inc.), and then quickly dried for 30 min in the antechamber of the glove box to remove the residual washing solvent. However, Li electrodes or Cu electrodes after cycling or CV premodulation were just immersed in the washing solvent overnight. To observe the morphology of lithium electrodeposits onto Cu electrode after CV premodulation, a thin polypropylene placer with a central hole was used instead of NPSS to eliminate impacts from separator and pressure. Surface investigation (morphology of lithium electrodeposits, element distribution) of the deposited Li or Cu electrodes were investigated using SEM (Hitachi S-4800) equipped with the energy dispersive X-ray analyser (Horiba, EX-350). Specimens were mounted onto specialized anti-contamination SEM holders and transferred into the antechamber of the SEM, where the flow of Ar gas is allowed during removal of the sample holder cover.

The surface chemistry of Li or Cu electrode was investigated by a Kratos ESCA-3400 spectrometer equipped with a monochromatized Mg X-ray source. The washed Li or Cu electrodes were dried and transferred into the XPS chamber using an in-house built stainless steel transfer holder. The binding energy was calibrated according to the hydrocarbon C 1s peak at 284.8 eV. The peaks were assigned based on previous reports and fitted using XPS Peak software. Raman spectra of the electrolytes were recorded using a spectrophotometer (Jobin-Yvon, T64000) with a 488 nm excitation laser to study the solvation state of each electrolyte. To prevent contamination from the air, the sample solution was placed into a quartz vial and tightly sealed in an Ar-filled glove box. The laser was radiated through a quartz crystal window.

Galvanostatic cycling measurements of Li|Li or Li|Cu cells were conducted with a battery testing system (Nagano, BTS-2004h) at 1.0–5.0 mA cm$^{-2}$ or a multichannel potentiostat-galvanostat (Biologic Science Instruments, VMP3) at 10 mA cm$^{-2}$ in an incubator of 25 °C. The coin-cell impedances were measured using a frequency response analyser (Solartron 1250) in the frequency range from 0.1 Hz to 1 MHz in a bench-top-type temperature chamber (Espec corp.) fixed at 25 °C. Z-plot software was employed for data analysis. Impedance evolution of Li|Li cells in different electrolytes with static aging time was recorded by VMP3 at 25 °C. Viscosity measurement of as-prepared electrolytes was conducted by a viscometer (Brookfield, DV-II) at ambient temperature.

The total energy for the formation of the stable-coordination structure between Li$^+$ and solvent in different molar ratios was computed by a MM2 calculation. Electric field distribution across planar lithium particles was simulated based on the finite element method.

**Electrochemical measurements.** CV premodulation: Coin cells of Li|Cu or *in situ* optical Li|Cu cells were first subjected to CVs in the potential of −0.3–3.0 V at the sweeping rate of 50 mV s$^{-1}$ for 5–30 scans using a Solartron 1285 potentiostat; coin cells of Li|Li underwent the CV programme in the potential range of −0.3–0.3 V at the sweeping rate of 10 mV s$^{-1}$ for 10 scans before cycling measurements and keep CV measurement every-half cycle in the potential range of −0.05–0.05 V at the same sweeping rate for five scans.

CE measurements of Li electrode in different electrolytes were performed by galvanostatic cycling of Li|Cu cells after CV premodulation at different current densities and different capacities with the cutoff potential of 0.5 V for lithium stripping. To standardize the testing, 70 μl of electrolyte was used in each coin cell. In this study, we used the following equation to calculate the average CE:

$$\text{Average Coulombic Efficiency} = \frac{\sum Q_{s,n}}{\sum Q_{p,n} + Q_r} \times 100\% \tag{1}$$

where $Q_{s,n}$ denotes the observed stripping capacity in the nth cycle, $Q_{p,n}$ the fixed plating capacity, $Q_r$ the capacity corresponding to the residual lithium electrodeposits after the CV premodulation.

*In situ* CV/QCM: The Ni (QA-A9M-NI) and Cu (QA-A9M-Cu) sputtered quartz crystal electrode was a 9 MHz AT-cut quartz (Seiko EG&G). The Ni (surface area 0.1963 cm$^2$) on the solution side was employed as the working electrode in a three-electrode cell system in connection with QCM (Seiko EG&G: PS-P800), where Li metal was used for the reference and counter electrode. The theoretical mass sensitivity was $9.57 \times 10^{-10}$ g Hz$^{-1}$. CV measurements were performed within a range of 0.0–3.1 V for LiFSI-2G4 and LiFSI-2G4-50 vol% DOL and

0.0–3.5 V for 1 M LiPF$_6$-EC-DEC at ambient temperature with a sweeping rate of 10 mV s$^{-1}$ in the glove box.

The ionic conductivity measurements for LiFSI-2G4-x vol% DOL mixtures were conducted with an AC conductivity method using a conventional conductivity cell with platinized platinum electrodes. Li|stainless steel coin cells were used to investigate the electrochemical window of as-prepared electrolytes at the sweeping rate of 10 mV s$^{-1}$ and 25 °C. The lithium ion transport number $t_{Li}^+$ was determined using symmetrical coin cells with a combination of AC impedance spectroscopy and the DC polarization method[44,45]. The cell potentials were measured after polarization for at least 4 h to obtain steady-state current data.

**Data availability.** The authors declare that the data supporting the findings of this study are available within the article and its Supplementary Information files. All other relevant data supporting the findings of this study are available on request.

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

## Acknowledgements

This work was supported by the Japan Science and Technology Agency under the programme of ALCA-SPRING: Specially Promoted Research for Innovative Next Generation Batteries. We thank Dr Tao Zhang, Takayuki Ichikawa, Mitsuhiro Matsumoto, Dr Hirokazu Okamura, Dr Peng Zhang, Dr Mengxuan Tang and Qi Chen for their assistance in conducting experiments. Valuable discussions from Dr Brian D. Adams are also greatly appreciated.

## Author contributions

H.W. and N.I. conceived and designed the experiments. H.W. prepared the electrolytes and performed the electrochemical, *in situ* observation, Raman spectroscopy measurements. H.W. and H.K. conducted XPS measurements. H.W. and H.S. carried out *in situ* CV/QCM measurements. H.W. and X.S. performed the theoretical computations and simulations. H.W., M.M., Y.M., Y.T., O.Y. and N.I. discussed the results and analysed the data. H.W. and N.I. wrote the manuscript.

## Additional information

**Competing interests:** The authors declare no competing financial interests.

