## [Peer Review File · Nature Communications]

Reviewers' comments:

Reviewer #1 (Remarks to the Author):

This manuscript reports on metallic lithium electrodes in a diluted solvate ionic liquid that allowed for stable cycling of Li electrodeposition/stripping at up to 12 mAh cm⁻². The key to achieving this was pretreatment of the Li electrode via a cyclic voltammetric protocol to generate a stable SEI layer where the concentration of DOL in the electrolyte was highly important. Overall the results are impressive and may be of practical benefit, however the following should be addressed

The manuscript is not written well. Admittedly there is a significant amount of information in this manuscript as evidenced by 22 supplementary figures and multi-component figures within the manuscript. Throughout the manuscript the discussion is truncated and not developed enough to clearly explain the key findings. I believe the manuscript may be more suited to a format that allows for a more expansive discussion of the topic and research findings.

On more technical aspects the following should be considered

In Figures 3b and 3c the origin of the peak at ca. 0.70 V should be explained.

Page 6 line 8: What does "the abnormal increase in the Coulomb was monitored.." mean?

For the discussion related to the EQCM data, the authors keep referring to the increase in mass of Ni? Ni is not being deposited in this experiment. Related to this work why didn't the authors use a Cu EQCM electrode to be consistent with the other experiments?

For the EIS experiments, the data needs to be fitted to an equivalent circuit model and quantitative information given for charge transfer resistance etc.

Some references to similarly related work should be included in the manuscript, for example

Adv. Sci. 2016, 1600168

Nature Communications, 2016, 7, 11794

ACS Applied Materials and Interfaces 2015, 7, 5950

Journal of Power Sources 228 (2013) 237

Overall this is a comprehensive and interesting piece of work but needs to be communicated much more effectively to get the key aspects of the study across.

Reviewer #2 (Remarks to the Author):

The authors report a reversible dendrite-free Li electrodes using diluted solvate ionic liquid with the robust solvation sheath to realize high areal-capacity and C.E. at ambient temperature. The idea is new and is worth to report. However, significant revision need to be done before its publication.

1. Fig. 2. 100% CE reported for many cells operated at different current densities and capacities is not correct. For example, the voltage profiles for Li stripping in Fig. 2c is only about 0.05 V instead of more than 0.5V. This means part of Li was not removed during the stripping process. On the other

hand, the stripping should be performed with the same cut off voltage instead of a variable voltage. The variable voltage means the authors used a current control instead of voltage control in the Li deposition/stripping process. In this case, the 100% CE is artificial value, not a real electrochemical behavior.

2. I don't understand the fluctuation of the data point shown in Fig. 2(d). The data shown in Fig. 2 indicates a poor Li cyclability. Vary low overvoltage ($\sim 0.04V$) and fluctuation of the voltage profile shown in Fig. 2 e and f may related to the soft short in the cell.

3. It states on page 6 that the QCM measurements were performed to study the SEI film formation. However, after reading that portion of the manuscript, I cannot find any evidence for an SEI at all. It seems like the authors prove underpotential deposition of Li onto Ni rather than any other film formation.

4. Fig .1, Wire growth instead of planetary growth. The surface area will grow with deposition so the current density will not be a constant.

5. Fig.1. caption, c) should be (e).

Reviewer #3 (Remarks to the Author):

Lithium metal-based batteries have been known as one of the most promising candidates for the next generation high-energy battery technologies. The reversible striping and plating of metallic lithium anode is critical for the safe use of these batteries. The author has reported a reversible dendritic-free metallic lithium electrode via CV premodulation and demonstrated its stable cycling in a dilute ionic liquid with high areal capacity (12 mAh cm^{-2}) and high current density (10 mA cm^{-2}). Dendrite-free technologies with high areal capacity have been rarely reported but very important for the application of lithium-metal batteries in large scale. If this study could be fully developed, it could be a breakthrough in the field of lithium metal batteries. This paper should be published after minor revision.

1. The author has studied the electrochemical performance of lithium metal electrode in Li/Li or Li/Cu cell. However, it will be more convincing if the author could provide its cycling performance in full cells, such as Li-S cells and Li-O₂ cells.

2. Typo: Page 3, Figure 1 caption: (c)↯(e); Space is missing in some place between unit and number and between "Fig." and number; Figure 2 caption: (f) is potential vs. capacity not voltage vs. time; Please give potential or voltage a reference electrode, such as V vs. Li/Li⁺; "Li/Li" or "Li|Li", please use the same symbol to indicate lithium-lithium cell.

3. Please provide the chemical composition of G4. Please also provide the full name of chemicals when they appear in the manuscript at the first time.

4. "Li₂O is the main constituent of the inner SEI film (7-55nm)", how did the author measure the distance to the surface? As far as I concern, we could know the etching time from the XPS analysis but the distance is very hard to obtain.

REVIEWERS' COMMENTS:

Reviewer #1 (Remarks to the Author):

The authors have satisfactorily addressed most of the comments, however there are still some issues to be resolved.

My original query on what the abnormal increase in the Coulomb means still does not make sense. This terminology needs to be replaced as it deals with charge per unit area and therefore cannot be referred to simply as the Coulomb.

As per my previous point on the Ni EQCM data, the mass of Ni is not increasing as Ni is not being deposited. A mass increase is observed due to either Li plating and/or SEI formation. This needs to be corrected.

The level of English needs to be significantly improved.

Reviewer #2 (Remarks to the Author):

Authors have measured the residual Li deposition during condition stage of the samples and introduced a revised formula to accurately define CE. They also provided reasonable explanations for several of other concerns I have before. I recommend the publication of this work in the current form.

Reviewer #3 (Remarks to the Author):

The authors have answered all the questions accordingly. And the paper should be accepted without further revision.

Reviewers' comments:

Reviewer #1 (Remarks to the Author):

Comment 1-1:

The manuscript is not written well. Admittedly there is a significant amount of information in this manuscript as evidenced by 22 supplementary figures and multi-component figures within the manuscript. Throughout the manuscript the discussion is truncated and not developed enough to clearly explain the key findings. I believe the manuscript may be more suited to a format that allows for a more expansive discussion of the topic and research findings.

Response 1-1:

We have rearranged our manuscript to emphasize on what key findings we want to unveil. Especially, we have rearranged the text in the section 2 of cycling ability of metallic lithium electrodes in the diluted solvate ionic liquid to show key findings other than simply list many supplementary experiments. New Figure 2 has been made. Moreover, we have added new discussion parts in the final section to correlate the observed experimental results with the underlying mechanism. Balance among each section has also been pursued.

On more technical aspects the following should be considered

Comment 1-2:

In Figures 3b and 3c the origin of the peak at ca. 0.70 V should be explained.

Response 1-2:

According to CVs of the three electrode cell of Li/Cu shown in new supplementary 14 a, the peak at ca. 0.70 V from Figure 3b and 3c should be associated with the part breakdown of the SEI formed at 0.5-0.60 V in the cathodic reduction scans.

Comment 1-3:

Page 6 line 8: What does "the abnormal increase in the Coulomb was monitored.."mean?

Response 1-3:

In the new Supplementary 12 a, during the cathodic sweep from 3.0- 0.6 V, the increase in the coulomb by 1 C cm^{-2} was observed. In general, the abrupt decrease in the coulomb was always witnessed during bulk lithium electrodeposition ($<0 \text{ V}$) in the cathodic sweep as shown in Figure 3a. Even if there is no lithium electrodeposition in the high potential range of 3.0-0.6 V during

initial sweeps, no change in the coulomb (i.e. a flat curve) was observed. However, after 30 sweeps, the Cu substrate had turned to positively-charged Cu₂Li substrate which is supposed to allow lithium stripping in the high potential range of 3.0-0.6 V regardless of the cathodic sweep or the anodic sweep.

Comment 1-4:

For the discussion related to the EQCM data, the authors keep referring to the increase in mass of Ni? Ni is not being deposited in this experiment. Related to this work why didn't the authors use a Cu EQCM electrode to be consistent with the other experiments?

Response 1-4:

When we started performing the EQCM measurements, only the Ni working electrode was available to use in our lab. To be consistent with our experiments, we also realized Cu sputtered quartz crystal electrode is the best candidate to study the dependence of the irreversible mass increase of Δm_{irr} (i.e. the SEI aging) on the sweeping. The new EQCM measurements using Cu working electrode is shown in Supplementary 14 a,b.

According to this Figure, lithium underpotential deposition (1.47 V)/stripping (2.58 V) was further demonstrated in this new three electrode cell. Two electrodes of Li/Cu cell also presented similar phenomenon. In addition, Δm_{irr} of Cu in LiFSI-2G4-50vol% DOL decreases with sweeping to be negligible when the sweep number reached 15 (in the case of Ni, the sweep number was 5), suggesting the presence of stable SEI on metallic lithium electrodes in contact with LiFSI-2G4-50vol% DOL during cycling measurements..

The main SEI of Li deposited on Cu arises from cathodic scans at ca. 0.6 V; the main SEI of Li deposited on Ni arises from cathodic at ca.0.3-0.4 V.

Comment 1-5:

For the EIS experiments, the data needs to be fitted to an equivalent circuit model and quantitative information given for charge transfer resistance etc.

Response 1-5:

We have added the equivalent circuit model to fit two typical impedance spectra in the supplementary Figure 15 a-b. Quantitative information for each component has also been included.

Comment 1-6:

Some references to similarly related work should be included in the manuscript, for example

- (1) Adv. Sci. 2016, 1600168
- (2) Nature Communications, 2016, 7, 11794
- (3) ACS Applied Materials and Interfaces 2015, 7, 5950
- (4) Journal of Power Sources 228 (2013) 237

Response 1-6:

We have added these four important literatures on metallic lithium anodes into corresponding places in the manuscript. The new assigned reference number is No 21 for (1), No 22 for (2), No 13 for (3) and No 12 for (4).

Reviewer #2 (Remarks to the Author):

The authors report a reversible dendrite-free Li electrodes using diluted solvate ionic liquid with the robust solvation sheath to realize high areal-capacity and C.E. at ambient temperature. The idea is new and is worth to report. However, significant revision need to be done before its publication.

Comment 2-1:

Fig. 2. 100% CE reported for many cells operated at different current densities and capacities is not correct. For example, the voltage profiles for Li stripping in Fig. 2c is only about 0.05 V instead of more than 0.5V. This means part of Li was not removed during the stripping process. On the other hand, the stripping should be performed with the same cut off voltage instead of a variable voltage. The variable voltage means the authors used a current control instead of voltage control in the Li deposition/stripping process. In this case, the 100% CE is artificial value, not a real electrochemical behavior.

Response 2-1:

We have revised the calculation method of Coulombic Efficiency and rearranged the second section. In contrast to conventional C.E. measurements, no CV-premodulation on Cu substrate was performed. As shown in new Figure 2a, the CV-premodulation not only can enhance the C.E. substantially but also drastically decrease the cell over potential for stripping.

According to our C.E. measurement protocol, there are two parameters for terminating the stripping process: (1) the same cut-off potential of 0.5 V (2) the theoretical stripping time for each fixed plating areal-capacity. The former condition was always fulfilled in advance in the case of cells having a C.E. lower than 100%. Even if the latter one was programmed, the C.E. measurements will not be terminated by the latter one when the C.E. is lower than 100%, i.e, the stripping time is consistently lower than the plating time at the same constant current density. However, when the apparent C.E. is gradually increased to 100% after many cycles in our experiments, the stripping cycle could be terminated at different voltages when the theoretical stripping time is reached.

In this study, a new equation was utilized to calculate the average C.E.:

$$\text{Average Coulombic Efficiency} = \frac{\sum Q_{s,n}}{\sum Q_{p,n} + Q_r} \times 100\%$$

where $Q_{s,n}$ denotes the stripping capacity in the n^{th} cycle, $Q_{p,n}$ -the fixed plating capacity, Q_r the initial capacity from residual lithium electrodeposits during CV-premodulation.

In order to get to know Q_r , the Li/Cu cell was subjected to stripping alone after the CV-premodulation and the Q_r was calculated as 0.25 mAh cm^{-2} as shown in the new supplementary Figure 8. Based on this new equation, the C.E. in our experiments is always lower than 100%. However, the new calculated C.E. are still exceptional in these harsh conditions.

In addition, the pre-existed residual lithium electrodeposits can drastically enhance the C.E. in Fig. 2a. Our C.E. measurement protocol always maintains the presence of residual lithium electrodeposits onto Cu substrate even the theoretical stripping was reached, which are pivotal for the generation of the integrated planar lithium particle to realize the complete subsequent stripping.

Comment 2-2:

2. I don't understand the fluctuation of the data point shown in Fig. 2(d). The data shown in Fig. 2 indicates a poor Li cyclability. Vary low overvoltage ($\sim 0.04\text{V}$) and fluctuation of the voltage profile shown in Fig. 2 e and f may related to the soft short in the cell.

Response 2-2:

Fluctuations in the previous Fig 2d are related to the cells cycled at low areal-capacity of 1.0 and 3.0 mAh cm^{-2} . As shown in the new Fig. 2b, the first C.E. strongly depends on the

initial plating areal-capacity of lithium electrodeposition onto Cu substrate. Figure 3d exhibits typical two plating plateaus on the pretreated Cu substrate. The first lower overpotential plateau should be associated with the lithium electrodeposition onto top of residual lithium electrodeposits in a growing fashion of dendrite-free two-dimensional large-planar lithium particles at higher positions (Fig. 3d); the higher overpotential plateau should be ascribed to the lithium electrodeposition onto pristine Cu surface directly or lithium surface at lower positions.

One can see the lithium stripping capacity is always higher than the lithium plating capacity originating from the first plateau, indicating that dendrite-free large-planar lithium particles are pivotal for complete lithium stripping. Moreover, the higher overpotential plateau regularly follows the lower overpotential plateau, suggesting two-dimensional layer-by-layer growth mode prevails in our experiments evidenced by the continuously increasing plating overpotential and otherwise three-dimensional lithium dendrite growth is expected to occur on the top surface of lithium electrodeposits directly with a shorter electrodeposition distance. The fraction of the second plateau for cells delivering high areal-capacity ($>5 \text{ mAh cm}^{-2}$) is so large that the lithium electrodeposition on Cu substrate at lower spots have sufficient time to grow till the coalescence with the top lithium electrodeposition layer within an unidirectional galvanostatic plating process. Meanwhile, lithium electrodeposits with an integrated planar morphology are supposed to be stripped completely. However, in the case of low-areal-capacity, the arrival time for the coalescence of these two layers should be substantially delayed (after 8 cycles for the cell delivering 3 mAh cm^{-2}) to result in the initial fluctuations of C.E.

The soft short of the cell usually represents the erratic potential fluctuation as well as the continuous potential decline with cycling due to the shortened interelectrode distance. However, in the symmetric Li|Li cells at 5.0 mA cm^{-2} (Fig. 2 e-f) and 10 mA cm^{-2} (Supplementary Fig. 11) exhibited similar potential evolution as shown in Li|Cu cells- the abrupt drop in cell potential due to the sudden change of lithium deposition spots from the top electrodeposited lithium into the bottom pristine lithium surface. As shown in the Fig. 4, symmetric Li|Li cells also underwent a drastic change during CV-premodulation. In the Fig. 4b, the overpotential for lithium plating/stripping at 5.0 mA cm^{-2} is found to be around 0.04 V after 10 sweeps owing to the drastic decrease in R_c . Both symmetric Li|Li cells at 5.0 mA cm^{-2} and 10 mA cm^{-2} presented quite steady cell potential evolutions upon cycling without spikes, erratic fluctuations and decreasing cell potential. In the supplementary Fig. 9, the absence of lithium dendrites on the cycled lithium metal surface further demonstrates the absence of the soft short of our cell.

Comment 2-3:

3. It states on page 6 that the QCM measurements were performed to study the SEI film formation. However, after reading that portion of the manuscript, I cannot find any evidence for an SEI at all. It seems like the authors prove underpotential deposition of Li onto Ni rather than any other film formation.

Response 2-3:

In this study, the QCM measurement was designed to monitor the mass change of working electrode of Ni or Cu, in the voltage of 0-3.1 V ahead of the onset of lithium bulk electrodeposition. The term of Δm_{irr} is referred to the accumulation of SEI onto working electrode upon sweeping. Negligible Δm_{irr} was only observed in the DSIL for Ni after 5 sweeps (Supplementary Fig. 13 d) and Cu for 15 sweeps (Supplementary Fig. 14 b), which could be acted as the solid evidence in support of the steady cell overpotential during the cycling tests of Fig. 3e-f and Supplementary Fig. 11.

Phenomenon of underpotential electrodeposition/stripping of lithium on Ni or Cu were surprisingly witnessed in the SIL and DSIL. Morphology of underpotential lithium electrodeposits were observed as the planar patches (new supplementary 14 d and f)--a favorable seed-morphology for two-dimensional layer-by-layer growth. This unexpected finding further demonstrates the importance of CV-premodulation.

Only based on QCM measurements, assignments of each cathodic reduction peak that corresponds to a SEI gradient are quite difficult to perform. We have revised some sentences on the QCM measurement in this section.

Comment 2-4:

4. Fig .1, Wire growth instead of planetary growth. The surface area will grow with deposition so the current density will not be a constant.

Response 2-4:

The usage of extra-thin Cu wire was intended to observe the evolution of lithium electrodeposition in a broad spatial visualization zone, which could act as a new convincing observation method of in-situ lithium electrodeposition. In these experiments, dendritic lithium electrodeposits with high surface area were observed in the case of conventional carbonate-based electrolytes. However, two-dimensional layer-by-layer lithium electrodeposits with low surface area were observed in the case of DSIL. The increase in the surface area on the planar substrate is also witnessed upon lithium electrodeposition (e.g. Figure 4c-d). The galvanostatic lithium electrodeposition is often conducted based on the geometric surface area of the substrate. In this study, the geometric surface area of Cu wire is in the range of 0.025—0.038 cm².

Comment 2-5:

5. Fig.1. caption, c) should be (e).

Response 2-5:

We have corrected this mistake.

Reviewer #3 (Remarks to the Author):

Lithium metal-based batteries have been known as one of the most promising candidates for the next generation high-energy battery technologies. The reversible stripping and plating of metallic lithium anode is critical for the safe use of these batteries. The author has reported a reversible dendritic-free metallic lithium electrode via CV premodulation and demonstrated its stable cycling in a dilute ionic liquid with high areal capacity (12 mAh cm^{-2}) and high current density (10 mA cm^{-2}). Dendrite-free technologies with high areal capacity have been rarely reported but very important for the application of lithium-metal batteries in large scale. If this study could be fully developed, it could be a breakthrough in the field of lithium metal batteries. This paper should be published after minor revision.

Comment 3-1:

1. The author has studied the electrochemical performance of lithium metal electrode in Li/Li or Li/Cu cell. However, it will be more convincing if the author could provide its cycling performance in full cells, such as Li-S cells and Li-O₂ cells.

Response 3-1:

It is worth applying the DSIL as the electrolyte for Li-S cells or Li-O₂ cells. In this manuscript, we are willing to report our strategy to stabilize metallic lithium anodes from fundamental aspects. Considering the sweeping-dependent electrochemical window of DSIL (Supplementary Fig. S3), we have begun the application work of DSIL in the Li-S cell, anode-free Li-ion batteries and hybrid Li-metal batteries. Related papers will be published in the near future.

Comment 3-2:

2. Typo: Page 3, Figure 1 caption: (c)∅(e); Space is missing in some place between unit and number and between "Fig." and number; Figure 2 caption: (f) is potential vs. capacity not voltage vs. time; Please give potential or voltage a reference electrode, such as V vs. Li/Li+; "Li/Li" or "Li|Li", please use the same symbol to indicate lithium-lithium cell.

Response 3-2:

We have corrected these mistakes including the captions, the necessary space and the unit. The same symbol of Li|Li has been utilized.

Comment 3-3:

3. Please provide the chemical composition of G4. Please also provide the full name of chemicals when they appear in the manuscript at the first time.

Response 3-3:

We have revised the end paragraph to add the information of involved chemicals in the introduction section.

Comment 3-4:

4. "Li₂O is the main constituent of the inner SEI film (7-55nm)", how did the author measure the distance to the surface? As far as I concern, we could know the etching time from the XPS analysis but the distance is very hard to obtain.

Response 3-4:

We used the reference etching rate for the SiO₂ based on the XPS instrument at our university. Considering the difference between the SEI ingredients and the SiO₂, all the discussion on the distance to the surface have been deleted.

Reviewer #1 (Remarks to the Author):

The authors have satisfactorily addressed most of the comments, however there are still some issues to be resolved.

Comment 1-1:

My original query on what the abnormal increase in the Coulomb means still does not make sense. This terminology needs to be replaced as it deals with charge per unit area and therefore cannot be referred to simply as the Coulomb.

Response 1-1:

We have replaced the terminology of the “*Coulomb ($C\text{ cm}^{-2}$)*” with “*Charge (Coulomb cm^{-2})*” shown in the main text, in Figure 3 and Supplementary Figure 12.

Comment 1-2:

As per my previous point on the Ni EQCM data, the mass of Ni is not increasing as Ni is not being deposited. A mass increase is observed due to either Li plating and/or SEI formation. This needs to be corrected.

Response 1-2:

We have corrected the phrase of “*the mass of Ni*” into “*the mass of the Ni substrate*” in the corresponding places.

Comment 1-3:

The level of English needs to be significantly improved.

Response 1-3:

A friend of ours, who is from Canada and have received his Ph.D. degree in Materials Chemistry, kindly assisted us in polishing our manuscript to get a higher accessibility.